# Adherence to COVID-19 policy measures: Behavioral insights from The Netherlands and Belgium

**Eline van den Broek-Altenburg** [1]✱*, **Adam Atherly** [2]✱

**1** Department of Radiology, Larner College of Medicine, University of Vermont, Burlington, Vermont, United States of America, **2** Department of Medicine, Larner College of Medicine, University of Vermont, Burlington, Vermont, United States of America

✱ These authors contributed equally to this work.
* Eline.Altenburg@med.uvm.edu

## Abstract

### Background

Since the start of the global COVID-19 pandemic, countries have been mirroring each other's policies to mitigate the spread of the virus. Whether current measures alone will lead to behavioral change such as social distancing, washing hands, and wearing a facemask is not well understood. The objective of this study is to better understand individual variation in behavioral responses to COVID-19 by exploring the influence of beliefs, motivations and policy measures on public health behaviors. We do so by comparing The Netherlands and Flanders, the Dutch speaking part of Belgium.

### Methods and findings

Our final sample included 2,637 respondents from The Netherlands and 1,678 from Flanders. The data was nationally representative along three dimensions: age, gender, and household income in both countries. Our key outcome variables of interest were beliefs about policy effectiveness; stated reasons for complying with public rules; and changes in behavior. For control variables, we included a number of measures of how severe the respondent believed Covid-19 to be and a number of negative side effects that the person may have experienced: loneliness, boredom, anxiety, and conflicts with friends and neighbors. Finally, we controlled for socio-demographic factors: age, gender, income (categorical), education (categorical) and the presence of Covid-19 risk factors (diabetes, high blood pressure, heart disease, asthma, allergies). The dependent variable for each of the estimation models is dichotomous, so we used Probit models to predict the probability of engaging in a given behavior.

We found that motivations, beliefs about the effectiveness of measures, and pre-pandemic behavior play an important role. The Dutch were more likely to wash their hands than the Flemish (15.4%, p<0.01), visit family (15.5%, p < .01), run errands (12.0%, p<0.05) or go to large closed spaces such as a shopping mall (21.2%, p<0.01). The Dutch were significantly less likely to wear a mask (87.6%, p<0.01). We also found that beliefs about the virus,

**Data Availability Statement:** We have shared our data from this study to a public repository as well as with a Supporting information file with this submission. The public repository can be found

here: https://github.com/ElinevandenBroek/Covid-19-NL-BE-data.

**Funding:** The authors received no specific funding for this work.

**Competing interests:** The authors have declared that no competing interests exist.

psychological effects of the virus, as well as pre-pandemic behavior play a role in adherence to recommendations.

## Conclusions

Our results suggest that policymakers should consider behavioral motivations specific to their country in their COVID-19 strategies. In addition, the belief that a policy is effective significantly increased the probability of the behavior, so policy measures should be accompanied by public health campaigns to increase adherence.

## Introduction

Since the global outbreak of COVID-19, policymakers and citizens have been debating which policy measures are effective at reducing the spread of the disease. Residents are being advised or mandated to socially distance themselves, wash their hands, and wear a facemask. However, making a measure mandatory does not guarantee conformance to the new set of rules.

One of the puzzles of COVID-19 is the lack of association between public health measures and COVID-19 outcomes. A recent review of the evidence found that relatively few public health measures, including rapid lockdowns, widespread testing and rapid boarder closures, had a measurable effect on COVID-19 caseload or mortality rates [1]. This raises the question of why country level restrictions were unrelated to country level outcomes. One possibility is that the reason is that behavioral responses *within* countries to similar policy initiatives varies sufficiently that linkages between policies and outcomes are statistically challenging to discern.

The objective of this study is to better understand individual variation in behavioral responses to COVID-19 by exploring the influence of beliefs, motivations and policy measures on public health behaviors. We evaluate the role of motivations, beliefs and pre-pandemic behavior in the policy response. We do so by comparing The Netherlands and Flanders, the Dutch speaking part of Belgium. These two regions are linguistically and economically similar, but employed different lockdown strategies between March and July 2020.

### Cross-country context

The Netherlands and Belgium are two relatively small neighboring countries in West Europe, both bordering the North Sea [2]. For this analysis, we focus on Flanders only, the Dutch-speaking part of Belgium. Apart from geographic location, the two regions are similar in many ways. The Netherlands is the sixth-largest economy in the European Union and plays an important role as a European transportation hub. It has a consistently high trade surplus, stable industrial relations, and low unemployment [2]. Flanders also has a highly developed transport network, which has helped develop a well-diversified economy, with a broad mix of transport, services, manufacturing, and high tech [2]. According to IndexMundi, a resource for country statistics, service and high-tech industries are mostly concentrated in the northern Flanders region [2].

GDP growth on both countries has been steady over the past years, with The Netherlands having a slightly higher GDP per capita than Belgium, but this includes the French speaking provinces of Belgium, which are poorer. Population growth is slightly higher in the whole country of Belgium [0.67%] than in The Netherlands [0.38%] but death rates have been very similar over the past years [9/1000 in Belgium versus 9.7 in The Netherlands]. Infant mortality

rates [3.4 vs. 3.5] as well as life expectancy at birth [81.2 vs. 81.5], both highly sensitive proxy measures of population health, are also very similar. Politically, both countries have a multi-party parliamentary system that requires the formation of coalitions to form government.

Despite the many similarities between the countries in geography, economics and language, there are important cultural differences between the two. Religiously, the two countries are different where more than half of Belgians are Roman Catholic and in The Netherlands, only 24% and more than half of Dutch people have no religion. There are also communication differences, such as in the perception of politeness [10]. In the Dutch cultural context, the focus is on the task to be achieved and being effective and the Flemish, despite using the same Dutch language, seem to reflect French where in the cultural context building the relationship is key [3,4]. These are just a few examples of where human behavior and interactions between the two countries differ.

## COVID-19 responses

Fig 1 shows the stringency in both government's responses since the beginning of the pandemic in March, 2020, using the Oxford COVID-19 Government Response Tracker [OxCGRT], which reports variation in government responses to COVID-19 across countries and time [5]. OxCGRT reports a "stringency" measure per country, developed by the Oxford Blavatnik School of Government, which has "systematically collected information on several different common policy responses that governments have taken to respond to the pandemic". It collects information on several different common policy responses that governments in 160 different countries have taken to respond to the pandemic and defines 17 indicators such as school closures and travel restrictions [5]. The OxCGRT combines the data from the different countries into a series of novel indices that aggregate various measures of government responses across dimensions of containment and closure; economic response; and health systems.

The countries introduced very similar measures regarding schools closing; work place closing; cancellation of public events; restrictions on gatherings with larger groups of people; domestic travel; international travel; public info campaigns; testing policies; contact tracing; income support as well as to some extent debt contract relief.

The most distinguishing policy was the "Stay home" order, which explains why the Belgian curve in Fig 1 falls under the Dutch curve in early June. The Dutch introduced a "recommendation" not to leave the house in March. The government never officially lifted this recommendation in June. The Belgian government introduced a "requirement" for not leaving house in March, with exceptions for daily exercise, grocery shopping, and 'essential' trips. This version of a "Stay Home" order was lifted on June 8, according to OxCGRT, moving to "no measures".

According to the Belgian Prime Minister Sophie Wilmes it was rather a key phase of deconfinement to allow Belgium to change their approach [6]. Concretely, it meant for the Flemish that, as of June 8, they were allowed to have close contact with 10 people per week, an "expanded personal bubble". Previously, they were allowed to meet with four other people from May 10 (Mother's day); but they could only interact with these four other people though. On June 10, the number expanded to 10 and the groups could vary. Group activities were also limited to 10 people, children included. Restaurants and bars could open, respecting social distancing; sporting activities all resumed and fitness centers were able to reopen. Religious worship also resumed, with up to 100 people in attendance. A week later, on June 15, the Belgian border also reopened to travel to and from other European countries [7,8].

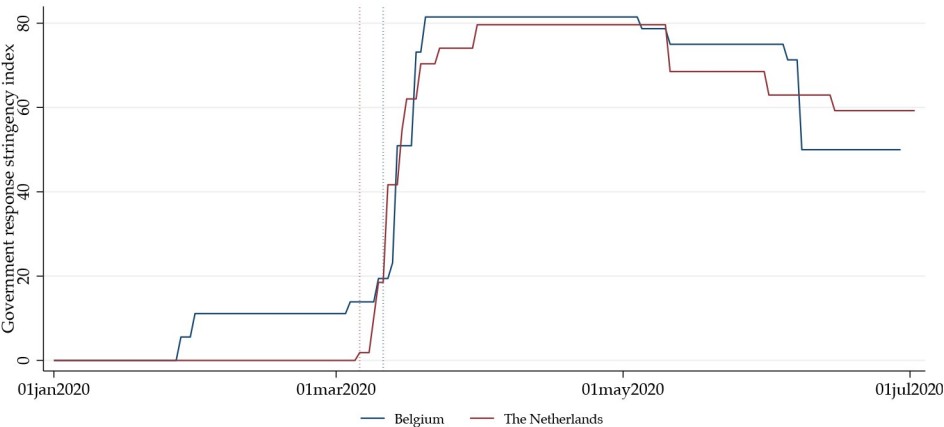

*Note*: Vertical dotted lines denote timing of first Covid-19 death in the country. Government response stringency index is computed by Hale et al. (2020).

**Fig 1. Stringency government response in The Netherlands and Belgium.**

**Lockdown effects on COVID-19 cases and deaths.** Analyzing the effect of the lockdowns on the number of cases, morbidity and mortality in each country has been challenging because the epidemic started earlier in some countries than others and other contributing factors are immutable [9,10]. For example, the infection fatality rate of COVID-19 can vary substantially across different locations and this may reflect differences in population age structure and case-mix of infected and deceased patients as well as multiple other factors. In addition, test capacity and testing policy have been moving targets in each country, making it difficult to compare prevalence rates between countries. Yet, studies have been massively using intuitive figures illustrating number of cases per country, reporting unreliable and misleading country comparisons of Covid-19 prevalence rates [11,12].

**Lockdown effects on economics.** Analyzing the effects of the lockdowns on economies is similarly challenging because policy measures have had varying effects on individual behaviors [13], individual socio-economic consequences [13–16] and global macroeconomic impact [17,18]. COVID-19 related measures have not only affected countries at different times in various ways, studies focusing on economic effects should also consider baseline differences. Analyzing economic effects should include other contributing factors, like the timing of policy measures, enforcement of the measures and the level of compliance and conformism in a country, which are partly culturally defined. The "stringency" and these cultural differences generally make it challenging to do country comparisons.

**Lockdown effects on individual behavior.** With the threat of a potential "second wave" of Covid-19 infections, countries are contemplating next steps after the phase of "lockdown". As mentioned, the word "lockdown" can refer to anything from mandatory geographic quarantines to non-mandatory recommendations to stay at home, closures of certain types of businesses, or bans on events and gatherings. Analyzing the effect of a lockdown on controlling the further spread of Covid-19 involves analyzing the effect of these individual policy measures on human behavior and interaction. Even though we do not yet know precisely how the virus is being spread [7,8], we do know that individual behavior and self-protecting behaviors have an effect on the pace of the spread.

The COVID-19 literature has already addressed how risk perception [19], fear [20,21], anxiety and suicidal thoughts [22] misinformation [23], (science) communication to the public [24], moral decision-making [24] =, leadership [25], psychosocial impacts [26], mental illness [27], and stress [28] affect human behavior and interaction. As far as we know, there is limited

evidence of the effect of policy measures themselves on behaviors and to what extent cultural context play a role in adherence to the policies, other than a study in Germany identifying factors affecting mask wearing [29]. Assuming that COVID-19 does spread primarily through human interaction, therefore, it is important to better understand these cultural factors affecting individual behavior.

## Materials and methods

This study has been reviewed and approved by the Institutional Review Board of the University of Vermont.

### Data

Our sample in the Netherlands consisted of approximately 3,000 respondents and 1,700 in Flanders. Upon excluding duplicate responses and those with missing data, our final dataset included 2,637 respondents from The Netherlands and 1,678 from Flanders. The data was nationally representative along three dimensions: age, gender, and household income in both countries.

The data were collected using quota sampling between June 2, 2020 and June 19, 2020 in the Netherlands and between May 29 and June 25 in Flanders with the support of market research companies Respondi in The Netherlands and Qualtrics in Flanders. The quota for both countries were based on statistics from Eurostat data in 2016. For Flanders, we used the overall population data for Belgium and divided them by the population for each age group. Before participating in the survey, respondents completed an online consent form. Participation was remunerated according to a general compensation schemes, defined by the survey companies. Respondents were excluded for completing the survey too quickly, which was defined as under 50% of the median response time.

### Analytic approach

Survey data collected information in a number of different domains. Our key outcome variables were self-reported individual behaviors. Respondents were asked how often they engaged in a series of different behaviors before, "soon after the Coivd-19 outbreak started in (their) country" and "now". Responses were given in a five-item attitudinal scale ranging from "Never" to "Always". For our analysis, we combined "Always" and "Very Often" into one group, with the reference group as "Never", "Rarely" and "Sometimes". This dichotomization reflects the policy goal of having individuals engage in the protective behaviors consistently.

We include eight different behaviors in our analysis. Three are personal/family: washing hands, wearing a mask and visiting family or friends. Five others reflect how the individual reacts to the broader community: participating in a social gathering with more than 10 people; going to a closed space like a supermarket; going to an open space such as a park; running errands like filling petrol or shopping in small shops and using public transportation.

Our key variables of interest for the first two models are beliefs about policy effectiveness. Respondents were asked how effective they believed each of the measures to be on a five-point attitudinal scale which ranged from "Not Effective at All" to "Extremely Effective". In each of the models, we included two measures that were not specific to any particular behavior to measure the influence of general belief about the efficacy of public response—fines for citizens who do not respect public safety measures and closing schools. We also included a measure of general attitude toward the effectiveness of government responses. Finally, for most models we had a belief specific to that behavior; i.e., beliefs about the value of requiring masks be worn was included in the model predicting mask wearing behavior. Similarly, a hand washing belief

variable was included in the hand washing behavior model, a belief in the value of shutting down public transportation was included in the public transportation measure, etc.

Our second two models examined stated reasons for complying with public rules. We included eight different potential reasons. These included protection of three different entities —self, family and the public, conformity ("Because everyone else did") and three different sources of recommendations—friends/family, doctors/public health officials, politicians. The final reasons were "legal restrictions".

Our third set of models looked at changes in behavior. We used the "now" timing of behavior and modeled changes from the time that Belgium lifted their stay at home order. We controlled for both country and timing, and looked at the interaction between the two to measure behavior changes.

For control variables, we included a number of measures of how severe the respondent believed Covid-19 to be. These included estimates of what fraction of people in the local area were infected, whether Covid-19 is less or more serious than influenza, the probability the person thinks they are or have been infected and the estimated probability someone with Covid-19 develops no symptoms and the probability they die. We also included a number of negative side effects that the person may have experienced: loneliness, boredom, anxiety, conflicts with friends and neighbors.

Finally, we controlled for socio-demographic factors: age, gender, income (categorical), education (categorical) and the presence of Covid-19 risk factors (diabetes, high blood pressure, heart disease, asthma, allergies). We also included a marker for whether the person reported having Covid-19 symptoms.

## Regression specification

For each of our static behaviors, we model the probability of engaging in the behavior as a function of the hypothesized effects plus control variables. The dependent variable for each of the models is dichotomous, so we used a Probit model to predict the probability of engaging in a given behavior. The basic specification for the belief models is:

$$y_i = \beta_0 + \beta_1 (\textit{Belief in Effectiveness}) + \beta_2 (\textit{Individual Risk}) + \beta_3 (\textit{Supportive of Public Policy}) + \beta_4 (\textit{COVID} - 19 \textit{ Risk}) + \beta_5 x + \varepsilon_i \tag{1}$$

Each of the terms in brackets represents a vector of variables. For $\beta_1$, we hypothesize that believing in the effectiveness of a specific activity (i.e. hand washing is effective) should be positively associated with engaging in the activity (washing hands) but should not be associated with other behaviors (e.g., wearing a mask). We include all beliefs in each model to test the counterfactual. For $\beta_2$, we include measures of age and presence of a significant COVID-19 risk factor (e.g., obesity, diabetes, asthma); we hypothesize that this coefficient should be positive if high-risk people are systematically more likely to engage in risk avoiding behavior. For $\beta_3$, we hypothesize that individuals who are generally supportive of public policy should be more likely to listen to recommendations. This concept is operationalized by a variable indicating support for public policy and by including a measure of support for closing schools and for public authorities levying fines for non-compliance, which are unrelated to any of the specific behaviors. Finally, we hypothesize that individuals who believe that COVID-19 is widespread in their community and deadly should be more likely to engage in protective behaviors (i.e., $\beta_4 > 0$). This concept is operationalized by questions asking beliefs about the percentage of individuals who have COVID-19 who die and who have no symptoms, whether COVID-19 is more or less severe than influenza, the fraction of the local population that has COVID-19 and beliefs about exposure to COVID-19.

Cross-national differences in this model are measured with the country indicator variable. X represents a vector of individual characteristics (income, education, rural) likely to be associated with beliefs.

The second set of models studies the influence of differences in motivations on behaviors. The basic specification for the motivation models is:

$$y_i = \beta_0 + \beta_1(Stated\ Motivation_i) + \beta_2(Country_i) + \beta_3(Stated\ Motivation_i * Country_i) + \beta_4 x_i + \varepsilon_i \qquad (2)$$

In this model, we examine the role of stated motivations on behaviors. We examine specifically the role of differences across countries in motivation and how that affects behaviors. In our model, The Netherlands = 1 and Belgium = 0. We hypothesize that individuals in The Netherlands will be more motivated by individual factors and less by desires for conformity.

To assess the influence of policies on behavior, we explicit differences in the timing of policy changes in The Netherlands and Belgium. Belgium relaxed their stay at home order on June 8, 2020, during the survey, which allows us to use a standard difference-in-differences model:

$$y_{ijt} = \beta_0 + \beta_1(Country_{ij}) + \beta_2(Post_{it}) + \beta_3(Country * Post_{ijt}) + \beta_4 x_i + \varepsilon_{ijt} \qquad (3)$$

In this model, time invariant differences between countries is shown by $\beta_1$, while $\beta_2$ controls for factors that effected both countries in the post period. The policy effect is given by $\beta_3$. The final vector, $x_i$, controls for individual factors that could confound the results.

We hypothesize that the influence of beliefs, individual risk and COVID-19 Risk should be consistent across private and public behaviors, while the influence of supportiveness of public policies and response to changes in public policies should be more significant for public than private behaviors.

Cultural differences were incorporated into the country indicator variable. This will include any country specific differences in behavioral adoption after controlling for demographics, COVID-19 risk, beliefs and other factors.

Coefficients were transformed into marginal probabilities, which are reported in the tables.

## Results and discussion

### Descriptive statistics

A total of 4,315 surveys were collected. Of these, 2,637 (61%) were from the Netherlands, and 1,678 (39%) were from Flanders. As mentioned, respondents were recruiting using quota sampling for age, gender and household income. Table 1 reports demographics; psychological influences; and agreement with government policies in the two countries.

Overall, we found that while many more Belgians experienced a drop in HHI and/or lost their jobs at least temporarily, there was much less agreement with government response among the Flemish than in The Netherlands. Psychologically, the Flemish reported more anxiety, loneliness, boredom and trouble sleeping than the Dutch. The Flemish were generally more likely to believe that people around them or they themselves may be infected with COVID-19.

### Beliefs and behaviors

Table 2 presents the results of the Probit regressions examining the relationship between beliefs and behaviors. Across all four measures of personal behavior—hand washing, mask wearing, visiting families and attending social gatherings—and there was a strong statistically significant country effect even after controlling for individual characteristics. Residents of the

**Table 1. Descriptive statistics.**

|  | Flanders | The Netherlands | Overall |
|---|---|---|---|
| Female* | 841 (50.12) | 1,409 (53.43) | 2,250 (52.14) |
| Age*** |  |  |  |
| 18–25 | 198 (11.80) | 285 (10.81) | 483 (11.19) |
| 26–35 | 304 (18.12) | 357 (13.54) | 661 (15.32) |
| 36–45 | 308 (18.36) | 382 (14.49) | 690 (15.99) |
| 46–55 | 335 (19.96) | 516 (19.57) | 851 (19.72) |
| 56–65 | 305 (18.18) | 461 (17.58) | 766 (17.75) |
| 65+ | 227 (13.53) | 636 (24.12) | 863 (20.00) |
| Income*** |  |  |  |
| First Quintile | 332 (19.79) | 402 (15.24) | 734 (17.01) |
| Second Quintile | 336 (20.02) | 511 (19.38) | 847 (19.63) |
| Third Quintile | 331 (19.73) | 574 (21.77) | 905 (20.97) |
| Fourth Quintile | 302 (18.00) | 373 (14.14) | 675 (15.64) |
| Fifth Quintile | 190 (11.32) | 220 (8.34) | 410 (9.50) |
| Prefer not to answer | 187 (11.14) | 557 (21.12) | 744 (17.24) |
| Education*** |  |  |  |
| Less than high school | 74 (4.41) | 57 (2.16) | 131 (3.04) |
| High school | 520 (30.00) | 1,157 (43,88) | 1,677 (38.86) |
| Some college | 126 (7.51) | 190 (7.21) | 316 (7.32) |
| Associate's degree | 250 (14.90) | 776 (29.43) | 1,026 (23.78) |
| Bachelor's | 402 (23.96) | 177 (6.71) | 579 (13.42) |
| Master's | 279 (16.63) | 238 (9.03) | 517 (11.98) |
| Doctorate/Professional degree | 27 (1.61) | 42 (1.59) | 69 (1.60) |
| Labor status before COVID-19*** |  |  |  |
| Full-time permanent position | 708 (42.19) | 707 (26.81) | 1,415 (32.79) |
| Full-time temporary | 36 (2.15) | 105 (3.98) | 141 (3.27) |
| Full-time self-employed | 59 (3.52) | 80 (3.03) | 139 (3.22) |
| Part-time permanent | 129 (7.69) | 415 (15.74) | 544 (12.61) |
| Part-time temporary | 25 (1.49) | 150 (5.69) | 175 (4.06) |
| Part-time self-employed | 14 (0.83) | 83 (3.15) | 97 (2.25) |
| Unemployed | 92 (5.48) | 119 (4.51) | 211 (4.89) |
| Out of labor force | 615 (36.65) | 978 (37.09) | 1,593 (36.92) |
| Lost Job Due to COVID-19*** |  |  |  |
| Yes, permanently | 42 (4.33) | 59 (3.83) | 101 (4.02) |
| Yes, temporarily | 190 (19.57) | 172 (11.17) | 362 (14.42) |
| No | 739 (76.11) | 1,309 (85.00) | 2,048 (81.56) |
| Experienced drop household income*** | 536 (31.94) | 470 (17.82) | 1,006 (23.31) |
| Covid-19 Symptoms yes*** | 776 (46.25) | 1,042 (39.51) | 1,818 (42.13) |
| Lonely due to policy changes*** | 510 (30.39) | 645 (24.46) | 1,155 (26.77) |
| Anxious due to policy changes*** | 621 (37.01) | 777 (29.47) | 1,398 (32.40) |
| Bored due to policy changes*** | 548 (32.66) | 741 (28.10) | 1,289 (29.87) |
| Belief % people around got infected** | 11.30 | 9.93 | 10.46 |
| Belief probability % you yourself got infected** | 16.00 | 13.86 | 14.69 |
| Support Gov't Policy*** |  |  |  |
| Strongly disagree | 148 (8.82) | 117 (4.44) | 265 (6.14) |
| Somewhat disagree | 255 (15.20) | 205 (7.77) | 460 (10.66) |
| Neither agree nor disagree | 417 (24.85) | 377 (14.30) | 794 (18.40) |

(*Continued*)

**Table 1.** (Continued)

| | Flanders | The Netherlands | Overall |
|---|---|---|---|
| Somewhat agree | 709 (42.25) | 1,112 (42.17) | 1,821 (42.20) |
| Strongly agree | 149 (8.88) | 826 (31.32) | 975 (22.60) |

Chi-square tests were used to analyze differences between the two countries.

*** p<0.01,

** p<0.05,

* p<0.1.

**Table 2. The effect of beliefs on private and public behavior.**

| Private Behaviors | Washing Hands | Wear Mask | Visit Family | Social 10+ |
|---|---|---|---|---|
| Netherlands | 0.154*** | -0.876*** | 0.155*** | 0.119 |
| | (0.0569) | (0.0591) | (0.0568) | (0.0778) |
| Belief Effective Washing Hands | 0.454*** | | | |
| | (0.0311) | | | |
| Belief Effective Requiring Masks | | 0.309*** | | |
| | | (0.0265) | | |
| Belief Effective banning gatherings | | | | -0.0848** |
| | | | | (0.0396) |
| Belief Effective of Closing Schools | 0.0531** | -0.0154 | -0.0483* | -0.0181 |
| | (0.0260) | (0.0267) | (0.0252) | (0.0358) |
| Belief Effectiveness of Fines | -0.0262 | -0.0444* | -0.0485** | 0.00520 |
| | (0.0239) | (0.0260) | (0.0230) | (0.0349) |
| Support Gov't Policy | -0.0140 | -0.0426 | -0.00881 | 0.0825** |
| | (0.0262) | (0.0260) | (0.0254) | (0.0363) |
| Public Behaviors | Shopping Mall | Parks | Run Errands | Public Transport |
| Netherlands | 0.212*** | -0.0343 | 0.120** | -0.103 |
| | (0.0521) | (0.0535) | (0.0476) | (0.0758) |
| Belief Effectiveness limiting mobility | -0.0721*** | -0.099*** | | |
| | (0.0253) | (0.0266) | | |
| Belief Eff. closing non-essential businesses | | | -0.0703*** | |
| | | | (0.0248) | |
| Belief Eff. shutting down public transport | | | | -0.185*** |
| | | | | (0.0380) |
| Belief in Effectiveness of Closing Schools | 0.00433 | -0.0517** | -0.00423 | 0.0407 |
| | (0.0240) | (0.0249) | (0.0229) | (0.0373) |
| Belief in Effectiveness of Fines | -0.0144 | 0.0259 | 0.0102 | 0.0302 |
| | (0.0221) | (0.0234) | (0.0199) | (0.0328) |
| Support Gov't Policy | -0.0488** | 0.0337 | -0.0353* | -0.0264 |
| | (0.0232) | (0.0247) | (0.0214) | (0.0347) |
| Observations | 4,311 | 4,311 | 4,311 | 4,311 |

Table reports marginal effects, standard errors in parentheses,

*** p<0.01,

** p<0.05,

* p<0.1. Also controlling for age, gender, income, education, living environment, covid-19 risk and beliefs. The column "social 10+" means gatherings of 10 or more people.

Netherlands were 15.4% more likely to wash their hands, but were far less likely to wear a mask and had a higher probability of visiting family.

None of the three personal behaviors that varied across country was associated with support of government policy, but believing the specific policy or recommendation to be effective significantly increased the probability of both washing hands (45.4%) and wearing masks (30.9%). Those who believed in the value of closing schools and fines were also less likely to visit family. Public behaviors were similar (Table 2). Residents of the Netherlands were more likely to visit large closed spaces (such as museum or shopping mall) and run errands, which was allowed in both countries at the time of this study. There were no differences in social gatherings—banned in both countries—or in visiting large open spaces such as parks or using public transportation—which was allowed in both countries.

Similarly, to private behavior, beliefs about the effectiveness of specific policies were strong predictors of behavior, including a negative association between all linked beliefs in policy effectiveness and behavior, including banning mass gatherings and socializing with ten or more people, limiting mobility and shopping/visiting parks, closing non-essential businesses and running errands and belief in the effectiveness of shutting public transportation and public transportation use. Supporters of government policies were less likely to socialize shop and run errands, but otherwise general policy support was not statistically significant.

On the control variables, age was generally insignificant, expect for a strong negative association with socializing with 10 or more individuals. Females were more likely to engage in all behaviors, while income was unrelated. Individuals that are more educated were less likely to shop, run errands or visit parks. None of age, income and education had consistent statistically significant effects, except that older persons were less likely to visit friends and wear masks. Those living outside urban areas were also less likely to wear masks. Beliefs regarding the prevalence of COVID-19 did predict behavior. The higher the believed fraction of the population that is infected, the more likely the respondent was to report mask wearing and, paradoxically, more likely to visit friends and family. Those who viewed COVID-19 as less risky were less likely to wear a mask, while the higher the percentage of people believed to die of COVID-19, the higher the probability of wearing masks and, again, of visiting family and friends. Anxiety played an important role as well; it was associated with a higher probability of washing hands and mask wearing.

The more severe people believed Covid-19, the less likely they were to engage in all behaviors, with the strongest influences for socializing with groups of 10+ (-0.21) and using public transportation (-0.17). Beliefs that Covid-19 was not serious (higher estimated percentage of people with no symptoms) increased the probability of going to gatherings of 10+ people and shopping. People who were lonely due to Covid-19 policy changes were more likely to visit parks and run errands.

## Reasons for behaviors

Turning to the stated reasons for behaviors, different behaviors had different motivations. Self-protection was predictive of all private behaviors (Table 3), with the strongest influence on mask wearing. Washing hands was also motivated by the desire to protect others—family and the public, while marking wearing was associated with conformity. Legal restrictions had the strongest influence on visiting family.

There were national differences in motivations. People in the Netherlands were less motivated by concern for family members to wash hands and they were also less likely to wear a mask for self-protection or based on politician's recommendations. Wearing masks among the Dutch was motivated by recommendations of friends or family.

**Table 3. The effect of motivations on behavior.**

| Private Behaviors | Washing Hands | Wear Mask | Visit Family | Social 10+ |
|---|---|---|---|---|
| Netherlands | 0.0599 | -0.81*** | 0.175 | 0.0196 |
| | (0.0991) | (0.124) | (0.107) | (0.141) |
| Changed Behavior | -0.135 | -0.140 | -0.161 | -0.250 |
| | (0.139) | (0.144) | (0.159) | (0.215) |
| Motivation: Self-Protection | 0.240** | 0.414*** | -0.215* | 0.191 |
| | (0.115) | (0.113) | (0.127) | (0.180) |
| Motivation: Protect Family | 0.516*** | 0.0700 | 0.0255 | -0.166 |
| | (0.101) | (0.0944) | (0.119) | (0.152) |
| Motivation: Protect Public | 0.207** | 0.0350 | -0.0853 | -0.0501 |
| | (0.0962) | (0.0774) | (0.101) | (0.135) |
| Motivation: Conformity | -0.139 | -0.32*** | 0.111 | 0.260 |
| | (0.119) | (0.107) | (0.128) | (0.163) |
| Motivation: Family/ Friends Recommend | -0.176 | 0.112 | 0.311** | 0.0469 |
| | (0.145) | (0.122) | (0.145) | (0.200) |
| Motivation: Physician Recommendation | 0.107 | -0.0340 | 0.0341 | -0.254* |
| | (0.0948) | (0.0783) | (0.101) | (0.139) |
| Motivation: Politian Recommendation | 0.135 | 0.0934 | -0.147 | -0.101 |
| | (0.110) | (0.0869) | (0.117) | (0.159) |
| Motivation: Legal Restrictions | 0.0746 | -0.0485 | -0.244** | -0.0987 |
| | (0.100) | (0.0821) | (0.110) | (0.148) |
| Netherlands*Changed Behavior | 0.175 | 0.237 | 0.0720 | 0.470* |
| | (0.178) | (0.214) | (0.194) | (0.259) |
| Netherlands*Motivation: Self-Protection | 0.0318 | -0.340** | 0.0152 | -0.462** |
| | (0.149) | (0.171) | (0.159) | (0.215) |
| Netherlands*Motivation: Protect Family | -0.321** | 0.0947 | 0.0188 | 0.0897 |
| | (0.132) | (0.145) | (0.147) | (0.188) |
| Netherlands*Motivation: Protect Public | -0.0129 | -0.187 | -0.170 | -0.117 |
| | (0.124) | (0.122) | (0.126) | (0.170) |
| Netherlands*Motivation: Conformity | -0.120 | 0.139 | -0.0324 | -0.282 |
| | (0.155) | (0.178) | (0.162) | (0.210) |
| NL*Motivation: Family/ Friends Recom. | 0.0580 | 0.335* | -0.0697 | 0.240 |
| | (0.194) | (0.189) | (0.190) | (0.253) |
| NL*Motivation: Physician Recommend | 0.136 | 0.0396 | -0.231* | -0.0229 |
| | (0.124) | (0.123) | (0.126) | (0.175) |
| NL*Motivation: Politician Recommend | -0.0664 | -0.47*** | 0.0397 | 0.188 |
| | (0.136) | (0.133) | (0.141) | (0.190) |
| Netherlands*Motivation: Legal | -0.0414 | -0.240* | 0.260* | 0.0979 |
| | (0.132) | (0.141) | (0.138) | (0.185) |
| Public Behaviors | Shopping Mall | Parks | Run Errands | Public Transport |
| Netherlands | 0.216** | 0.102 | 0.186* | -0.0911 |
| | (0.103) | (0.112) | (0.0963) | (0.143) |
| Changed Behavior | -0.00195 | -0.0274 | -0.119 | -0.167 |
| | (0.146) | (0.154) | (0.135) | (0.192) |
| Motivation: Self-Protection | -0.111 | 0.0676 | -0.0919 | 0.0963 |
| | (0.118) | (0.122) | (0.108) | (0.157) |
| Motivation: Protect Family | -0.144 | -0.0514 | 0.0135 | -0.286** |
| | (0.103) | (0.104) | (0.0952) | (0.133) |

*(Continued)*

**Table 3.** (Continued)

| | | | | |
|---|---|---|---|---|
| Motivation: Protect Public | 0.0281 | 0.0661 | 0.0500 | -0.0451 |
| | (0.0916) | (0.0881) | (0.0806) | (0.122) |
| Motivation: Conformity | 0.274** | 0.0536 | 0.0696 | 0.272* |
| | (0.113) | (0.115) | (0.106) | (0.150) |
| Motivation: Family/ Friends Recommend | 0.117 | 0.0353 | 0.203 | 0.334** |
| | (0.138) | (0.137) | (0.125) | (0.168) |
| Motivation: Physician Recommendation | -0.160* | -0.0199 | -0.105 | -0.132 |
| | (0.0915) | (0.0883) | (0.0811) | (0.125) |
| Motivation: Politian Recommendation | -0.0938 | -0.0442 | -0.173* | -0.255* |
| | (0.104) | (0.0986) | (0.0926) | (0.147) |
| Motivation: Legal Restrictions | -0.0214 | 0.0371 | -0.0131 | -0.229* |
| | (0.0953) | (0.0914) | (0.0847) | (0.137) |
| Netherlands*Changed Behavior | -0.0433 | 0.00283 | 0.0445 | 0.214 |
| | (0.179) | (0.191) | (0.168) | (0.250) |
| Netherlands*Motivation: Self-Protection | -0.149 | -0.175 | -0.0801 | -0.220 |
| | (0.146) | (0.154) | (0.136) | (0.207) |
| Netherlands*Motivation: Protect Family | 0.113 | -0.000247 | 0.0106 | 0.179 |
| | (0.128) | (0.132) | (0.119) | (0.181) |
| Netherlands*Motivation: Protect Public | -0.109 | 0.0450 | -0.187* | -0.164 |
| | (0.113) | (0.114) | (0.102) | (0.169) |
| NL*Motivation: Conformity | -0.231 | -0.0394 | -0.0175 | -0.0974 |
| | (0.145) | (0.150) | (0.136) | (0.210) |
| NL*Motivation: Family/ Friends Recom. | -0.0391 | 0.0531 | -0.218 | -0.0234 |
| | (0.178) | (0.182) | (0.166) | (0.247) |
| NL*Motivation: Physician Recommend | 0.188* | 0.00148 | -0.0654 | -0.291 |
| | (0.114) | (0.114) | (0.103) | (0.178) |
| NL*Motivation: Politician Recommend | 0.0452 | -0.0232 | 0.106 | -0.0986 |
| | (0.124) | (0.123) | (0.112) | (0.196) |
| Netherlands*Motivation: Legal | 0.112 | -0.0502 | 0.0896 | 0.190 |
| | (0.120) | (0.121) | (0.109) | (0.189) |
| Observations | 4,311 | 4,311 | 4,311 | 4,311 |

Table reports marginal effects, standard errors in parentheses,

*** p<0.01,

** p<0.05,

* p<0.1. Also controlling for age, gender, income, education, living environment, covid-19 risk and beliefs. The column "social 10+" means gatherings of 10 or more people.

For public behaviors, conformity was more important and self-protection less. Conformity motivated not shopping and not using public transportation, while self-protection was not statistically significantly related to any behavior. For the Netherlands, people motivated by self-protection were less likely to socialize while those motivated by concerns about the public were less likely to run errands and those motivated by physicians deferred shopping.

## Effect of policy change on behavior

One factor that was not associated with changes in behavior was actual policy changes (Table 4). None of the behaviors changed differentially in Flanders in response to Belgium lifting the stay at home order. There was a time effect for many of the variables—during the time

**Table 4. The effect of policy changes on behavior (Private).**

|  | Washing Hands | Wear Mark | Visit Family | Social 10+ |
|---|---|---|---|---|
| Netherlands | 0.0303 | -1.207*** | 0.112 | 0.167 |
|  | (0.0737) | (0.0705) | (0.0772) | (0.119) |
| Post Time Period | -0.0351 | -0.0578 | 0.178** | 0.305** |
|  | (0.0792) | (0.0670) | (0.0815) | (0.119) |
| Flanders*Post-lockdown | -0.0997 | 0.132 | -0.125 | -0.137 |
|  | (0.0996) | (0.0963) | (0.102) | (0.150) |
|  | Shopping | Parks | Run Errands | Public Transport |
| Netherlands | 0.0608 | -0.00728 | 0.0938 | -0.324*** |
|  | (0.0687) | (0.0711) | (0.0633) | (0.0919) |
| Post Time Period | -0.0601 | 0.0392 | 0.122* | -0.128 |
|  | (0.0764) | (0.0763) | (0.0693) | (0.0921) |
| Flanders*post-lockdown | 0.127 | -0.00454 | -0.0566 | 0.149 |
|  | (0.0942) | (0.0967) | (0.0859) | (0.125) |
| Observations | 4,311 | 4,311 | 4,311 | 4,311 |

Table reports marginal effects, standard errors in parentheses,

*** $p < 0.01$,

** $p < 0.05$,

* $p < 0.1$. Also controlling for age, gender, income, education. The column "social 10+" means gatherings of 10 or more people.

when Belgium lifted their restrictions, respondents were more likely to visit family, socialize with 10+ people and run errands. However, that influence was consistent across both countries.

## Discussion

In this paper, we explored the influence of policy measures and beliefs about their effectiveness on public health behaviors and evaluate if motivations, beliefs and pre-pandemic behavior have played a modifying role in the policy impact. We did so by comparing two countries, The Netherlands and Flanders, with two different lockdown strategies between March and July 2020. Our results suggest that the effectiveness of policies will depend to some extent on the public beliefs about the effectiveness of the policies.

In both countries, the belief that a policy is effective significantly increased the probability of the behavior. For example, about a third who believed that putting a policy in place for wearing facemasks is effective and reported wearing a mask. For all measures of personal behavior (washing hands, wearing mask, and visit family and friends) and some "public" behaviors (visit large closed spaces and running errands), there was a strong statistically significant country effect even after controlling for individual characteristics. Country itself may not necessarily be representative of culture, but beliefs about the pandemic, beliefs about whether or not policy measures are effective, pre-pandemic behavior and reasons for behaviors together show a cultural influence. In some cases, the country influence may have outweighed the influence of the individual beliefs. For example, people in The Netherlands were 22 percentage points more likely to visit a mall; individual belief in the value on restricting such visits reduced the likelihood of visits by 7 percentage points. Thus, in this case, the culture coefficient is greater than the estimated effect of individual beliefs. This suggests that policies to change behavior starts from a cultural "baseline" that will make policy implementation much more difficult in some places than others.

We found important differences in health behaviors in the two countries. Where the Flemish were more likely to engage in self-protecting behaviors to follow conformity and to protect the public and their families, the Dutch were more likely to engage in these behaviors to protect themselves. In both countries, people were more likely to decrease visits to friends and family following legal measures, implying that fines are an effective way to motivate citizens in public behaviors, regardless of beliefs.

Beliefs regarding the prevalence of COVID-19 did predict behavior. The higher the believed fraction of the population that is infected, the more likely the respondent was to report mask wearing. This implies that the role of fear, and the beliefs/perceptions about COVID-19 prevalence, are important factors affecting self-protecting behaviors.

## Limitations

One limitation in any survey is that the questions and behaviors presented must be realistic. Therefore, it is important to establish a feasible set of behaviors and exclude the potentially implausible ones. Others have used psychometric testing of scales standardized instruments for assessing all the behaviors assessing all the behaviors, however, we focused exclusively on those measured in the Oxford response tracker. Also, any sort of omitted variable bias can be a significant problem in a cross-country comparisons of this kind. If a variable that helps explain behaviors is correlated with any of the regressors and is not included in the regression, then coefficient estimates and standard will be biased. Therefore, we focused on interpreting results by country, without drawing strong cross-country conclusions.

## Conclusions

There are important behavioral differences between countries. Therefore, it is not recommended to adopt "one-size-fits-all" policy strategies to mitigate the spread of COVID-19. Any country seeking to change its citizens' behaviors must incorporate education and communication to change beliefs and motivations. It is important to improve public health campaigns providing citizens with information regarding risks and potential remedies. However, it is important to keep in mind that pre-pandemic behavior will not change easily: countries' culture and context matter when designing strategies to mitigate the spread of COVID-19. These findings provide guidance to policymakers who seek to mirror other countries "best practices". Whatever policy measures governments take in the months to come; it is important to realize that policies that are most impactful in one circumstance will not necessarily be equally effective in another.

## Supporting information

**S1 File.**
(CSV)

**S2 File.**
(XLSX)

## Acknowledgments

We thank all of the study participants for their time and effort. We are grateful to Michele Belot and Egon Tripodi for their generous support in the data collection process and for insightful discussion and contributions to the drafting of this manuscript. We thank the UVM

Research Protections Office and Institutional Review Board for rapid turnaround of COVID-19-related projects.

## Author Contributions

**Conceptualization:** Eline van den Broek-Altenburg, Adam Atherly.

**Data curation:** Eline van den Broek-Altenburg.

**Formal analysis:** Eline van den Broek-Altenburg, Adam Atherly.

**Methodology:** Eline van den Broek-Altenburg, Adam Atherly.

**Visualization:** Eline van den Broek-Altenburg.

**Writing – original draft:** Eline van den Broek-Altenburg, Adam Atherly.

**Writing – review & editing:** Eline van den Broek-Altenburg.

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
