## [Decision Letter · Decision Letter 0]

3 Feb 2021

PONE-D-20-40617

Adherence to COVID-19 Policy Measures:

Behavioral Insights from The Netherlands and Belgium

PLOS ONE

Dear Dr. van den Broek-Altenburg,

Thank you for submitting your manuscript to PLOS ONE. After careful consideration, we feel that it has merit but does not fully meet PLOS ONE’s publication criteria as it currently stands. Therefore, we invite you to submit a revised version of the manuscript that addresses the points raised during the review process.

We look forward to receiving your revised manuscript.

Kind regards,

Amir H. Pakpour, Ph.D.

Academic Editor

PLOS ONE

Journal Requirements:

4. Please include a copy of Tables 6 and 7 which you refer to in your text on page 20.

Reviewers' comments:

Reviewer's Responses to Questions

**Comments to the Author**

1. Is the manuscript technically sound, and do the data support the conclusions?

Reviewer #1: Partly

2. Has the statistical analysis been performed appropriately and rigorously? 

Reviewer #1: Yes

3. Have the authors made all data underlying the findings in their manuscript fully available?

Reviewer #1: Yes

4. Is the manuscript presented in an intelligible fashion and written in standard English?

Reviewer #1: Yes

5. Review Comments to the Author

Reviewer #1: The study entitled “Adherence to COVID-19 Policy Measures: Behavioral Insights from The Netherlands and Belgium” investigated the individual variation in adhering the behavioral recommendations made by the government during the COVID-19 pandemic. The roles of motivations, beliefs and pre-pandemic behavior in the policy response were evaluated. The findings indicate that the citizens in Netherlands and those in Belgium had different attitudes and behaviors reacting to the COVID-19. Therefore, the authors suggested that it is not recommended to adopt “one-size-fits-all” policy strategies to mitigate the spread of COVID-19. The manuscript tackles an important and timely research question with the following strengths: (1) a large sample size; (2) cross-country comparison; and (3) rigorous statistical methods. However, several issues should be remediated before I recommend publication. Please see my specific comments below.

1. The Abstract is not in a good shape. I think that Plos One allows much more words than the current presented Abstract. The authors are thus needed to provide more details and findings (with statistical values) in the Abstract. I would also prefer to see a structured Abstract as this is clear for readers to have a full picture in the beginning of reading this paper.

2. The authors do not review the literature thoroughly. Specifically, many papers on policy, behaviors, and associated factors during COVID-19 pandemic have been published. Therefore, the authors should take references from them. Below are some of my suggestions for the authors to use and strengthen their literature review.

Olatunji OS, Ayandele O, Ashirudeen D, Olaniru OS. “Infodemic” in a pandemic: COVID-19 conspiracy theories in an african country. Soc Health Behav 2020;3:152-7

Lin MW, Cheng Y. Policy actions to alleviate psychosocial impacts of COVID-19 pandemic: Experiences from Taiwan. Soc Health Behav 2020;3:72-3

Rieger MO. To wear or not to wear? Factors influencing wearing face masks in Germany during the COVID-19 pandemic. Soc Health Behav 2020;3:50-4

Chang KC, Strong C, Pakpour AH, Griffiths MD, Lin CY. Factors related to preventive COVID-19 infection behaviors among people with mental illness. J Formos Med Assoc. 2020;119(12):1772-1780. doi:10.1016/j.jfma.2020.07.032

Pramukti I, Strong C, Sitthimongkol Y, Setiawan A, Pandin MGR, Yen CF, Lin CY, Griffiths MD, Ko NY. Anxiety and Suicidal Thoughts During the COVID-19 Pandemic: Cross-Country Comparative Study Among Indonesian, Taiwanese, and Thai University Students. J Med Internet Res 2020;22(12):e24487

3. I cannot understand why the authors reported t-values for categorical data on Table 1. Also, the meaning of the asterisks is not provided for Table 1.

4. The meaning of social 10+ is unclear for Tables 2 to 4.

5. The study should have a limitation section to remind the readers, especially I found that the study did not use standardized instruments for assessing all the behaviors (e.g., Chang et al. have validated behavioral instruments during COVID-19). Therefore, it is unclear whether the measures used in the present study are robust.

Chang KC, Hou WL, Pakpour AH, Lin CY, Griffiths MD. Psychometric Testing of Three COVID-19-Related Scales Among People with Mental Illness [published online ahead of print, 2020 Jul 11]. Int J Ment Health Addict. 2020;1-13. doi:10.1007/s11469-020-00361-6

6. PLOS authors have the option to publish the peer review history of their article (what does this mean?). If published, this will include your full peer review and any attached files.

Reviewer #1: No

---

## [Author Response · Author response to Decision Letter 0]

28 Mar 2021

https://journals.plos.org/plosone/s/file?id=wjVg/PLOSOne_formatting_sample_main_body.pdf andhttps://journals.plos.org/plosone/s/file?id=ba62/PLOSOne_formatting_sample_title_authors_affiliations.pdf

We edited the manuscript and title page according to the PLOS ONE style templates. 

We had our manuscript copyedited at the University of Vermont.

We have now shared our data from this study to a public repository as well as with a supporting information file with this submission. Please update the Data Availability statement on our behalf. The public repository can be found here: https://github.com/ElinevandenBroek/Covid-19-NL-BE-data

4. Please include a copy of Tables 6 and 7 which you refer to in your text on page 20.

This was a typo, this paragraph refers to table 4. We edited the text.

We do not have supporting information files in our revision, other than the data from this study which will not be included in the in-text citations. 

 

Reviewer #1: The study entitled “Adherence to COVID-19 Policy Measures: Behavioral Insights from The Netherlands and Belgium” investigated the individual variation in adhering the behavioral recommendations made by the government during the COVID-19 pandemic. The roles of motivations, beliefs and pre-pandemic behavior in the policy response were evaluated. The findings indicate that the citizens in Netherlands and those in Belgium had different attitudes and behaviors reacting to the COVID-19. Therefore, the authors suggested that it is not recommended to adopt “one-size-fits-all” policy strategies to mitigate the spread of COVID-19. The manuscript tackles an important and timely research question with the following strengths: (1) a large sample size; (2) cross-country comparison; and (3) rigorous statistical methods. However, several issues should be remediated before I recommend publication. Please see my specific comments below.

1. The Abstract is not in a good shape. I think that Plos One allows much more words than the current presented Abstract. The authors are thus needed to provide more details and findings (with statistical values) in the Abstract. I would also prefer to see a structured Abstract as this is clear for readers to have a full picture in the beginning of reading this paper.

Thank you very much for your suggestion. We have rewritten the abstract in a structured form and added more details and findings on pages 1 and 2.

2. The authors do not review the literature thoroughly. Specifically, many papers on policy, behaviors, and associated factors during COVID-19 pandemic have been published. Therefore, the authors should take references from them. Below are some of my suggestions for the authors to use and strengthen their literature review.

We appreciate the feedback and have added a more thorough review of the literature, including the reviewer’s suggestions below.

Olatunji OS, Ayandele O, Ashirudeen D, Olaniru OS. “Infodemic” in a pandemic: COVID-19 conspiracy theories in an african country. Soc Health Behav 2020;3:152-7

Lin MW, Cheng Y. Policy actions to alleviate psychosocial impacts of COVID-19 pandemic: Experiences from Taiwan. Soc Health Behav 2020;3:72-3

Rieger MO. To wear or not to wear? Factors influencing wearing face masks in Germany during the COVID-19 pandemic. Soc Health Behav 2020;3:50-4

Chang KC, Strong C, Pakpour AH, Griffiths MD, Lin CY. Factors related to preventive COVID-19 infection behaviors among people with mental illness. J Formos Med Assoc. 2020;119(12):1772-1780. doi:10.1016/j.jfma.2020.07.032

Pramukti I, Strong C, Sitthimongkol Y, Setiawan A, Pandin MGR, Yen CF, Lin CY, Griffiths MD, Ko NY. Anxiety and Suicidal Thoughts During the COVID-19 Pandemic: Cross-Country Comparative Study Among Indonesian, Taiwanese, and Thai University Students. J Med Internet Res 2020;22(12):e24487

3. I cannot understand why the authors reported t-values for categorical data on Table 1. Also, the meaning of the asterisks is not provided for Table 1.

We have added caption for Table 1, explaining that the asterisks in error and deleted those from the table. 

4. The meaning of social 10+ is unclear for Tables 2 to 4.

This means social gatherings of 10 or more people. We added this caption below the tables 2 tot 4. 

5. The study should have a limitation section to remind the readers, especially I found that the study did not use standardized instruments for assessing all the behaviors (e.g., Chang et al. have validated behavioral instruments during COVID-19). Therefore, it is unclear whether the measures used in the present study are robust.

Thank you for this suggestion. We have added a Limitation section after the Discussion section on page 24. We have addressed this issue explaining why we used these methods not using standardized instruments, citing Change et al. and added text.

Chang KC, Hou WL, Pakpour AH, Lin CY, Griffiths MD. Psychometric Testing of Three COVID-19-Related Scales Among People with Mental Illness [published online ahead of print, 2020 Jul 11]. Int J Ment Health Addict. 2020;1-13. doi:10.1007/s11469-020-00361-6

---

## [Decision Letter · Decision Letter 1]

5 Apr 2021

Adherence to COVID-19 Policy Measures:

Behavioral Insights from The Netherlands and Belgium

PONE-D-20-40617R1

Dear Dr. van den Broek-Altenburg,

We’re pleased to inform you that your manuscript has been judged scientifically suitable for publication and will be formally accepted for publication once it meets all outstanding technical requirements.

Kind regards,

Amir H. Pakpour, Ph.D.

Academic Editor

PLOS ONE

Additional Editor Comments (optional):

Reviewers' comments:

Reviewer's Responses to Questions

**Comments to the Author**

1. If the authors have adequately addressed your comments raised in a previous round of review and you feel that this manuscript is now acceptable for publication, you may indicate that here to bypass the “Comments to the Author” section, enter your conflict of interest statement in the “Confidential to Editor” section, and submit your "Accept" recommendation.

Reviewer #1: All comments have been addressed

2. Is the manuscript technically sound, and do the data support the conclusions?

Reviewer #1: Yes

3. Has the statistical analysis been performed appropriately and rigorously? 

Reviewer #1: Yes

4. Have the authors made all data underlying the findings in their manuscript fully available?

Reviewer #1: Yes

5. Is the manuscript presented in an intelligible fashion and written in standard English?

Reviewer #1: Yes

6. Review Comments to the Author

Reviewer #1: The authors have satisfactorily addressed all my prior comments. I am happy to see this revision and have no more comments.

7. PLOS authors have the option to publish the peer review history of their article (what does this mean?). If published, this will include your full peer review and any attached files.

Reviewer #1: No

---

## [Editor Report · Acceptance letter]

23 Apr 2021

PONE-D-20-40617R1 

Adherence to COVID-19 Policy Measures:Behavioral Insights from The Netherlands and Belgium  

Dear Dr. van den Broek-Altenburg:

I'm pleased to inform you that your manuscript has been deemed suitable for publication in PLOS ONE. Congratulations! Your manuscript is now with our production department. 

Kind regards, 

on behalf of

Dr. Amir H. Pakpour 

Academic Editor

PLOS ONE